# Inhibitors against DNA Polymerase I Family of Enzymes: Novel Targets and Opportunities

**DOI:** 10.3390/biology13040204

**Published:** 2024-03-22

**Authors:** Saathvik Kannan, Samuel W. Gillespie, Wendy L. Picking, William D. Picking, Christian L. Lorson, Kamal Singh

**Affiliations:** 1Bond Life Sciences Center, University of Missouri, Columbia, MO 65211, USA; skannan@missouri.edu (S.K.); sgillespie@missouri.edu (S.W.G.); pickingw@missouri.edu (W.L.P.); wendy.picking@missouri.edu (W.D.P.); lorsonc@missouri.edu (C.L.L.); 2Department of Veterinary Pathobiology, University of Missouri, Columbia, MO 65211, USA

**Keywords:** DNA polymerase I, Polymerase θ, reverse transcriptase, homologous recombination, antibiotics, apicoplast

## Abstract

**Simple Summary:**

DNA polymerases are essential enzymes for the growth and survival of a cell. These enzymes replicate and/or repair the cellular genome. Some DNA polymerases have been associated with specific diseases. For example, Y-family DNA polymerases have been associated with cancer. Family A polymerases (typified by *E. coli* DNA polymerase I) have been extensively studied, primarily to understand the mechanism of DNA replication. Human mitochondrial DNA polymerase γ and DNA polymerase θ are exceptions as they have roles in genetic diseases and cancer, respectively. However, recent studies have uncovered novel roles of Family A polymerases. Thus, polymerase θ has been shown to conduct reverse transcriptase activity, an activity displayed by retroviruses’ (such as HIV) reverse transcriptases. The association of these polymerases with diseases makes them attractive therapeutic targets against diseases such as cancer and bacterial/parasitic infections. This perspective is focused on Family A polymerases as potential therapeutic targets for the development of new classes of antibiotics, antimalarial, and anticancer drugs. These new drugs could be used either alone or in combination with current interventions to circumvent drug-resistance. Modern techniques, such as artificial intelligence and docking of millions of drug-like compounds, can expedite the discovery of new interventions targeting Family A polymerases.

**Abstract:**

DNA polymerases replicate cellular genomes and/or participate in the maintenance of genome integrity. DNA polymerases sharing high sequence homology with *E. coli* DNA polymerase I (pol I) have been grouped in Family A. Pol I participates in Okazaki fragment maturation and in bacterial genome repair. Since its discovery in 1956, pol I has been extensively studied, primarily to gain deeper insights into the mechanism of DNA replication. As research on DNA polymerases advances, many novel functions of this group of polymerases are being uncovered. For example, human DNA polymerase θ (a Family A DNA pol) has been shown to synthesize DNA using RNA as a template, a function typically attributed to retroviral reverse transcriptase. Increased interest in drug discovery against pol θ has emerged due to its roles in cancer. Likewise, Pol I family enzymes also appear attractive as drug-development targets against microbial infections. Development of antimalarial compounds targeting apicoplast apPOL, an ortholog of Pol I, further extends the targeting of this family of enzymes. Here, we summarize reported drug-development efforts against Family A polymerases and future perspective regarding these enzymes as antibiotic targets. Recently developed techniques, such as artificial intelligence, can be used to facilitate the development of new drugs.

## 1. Introduction

DNA replication and repair are essential for the propagation of all forms of life. DNA polymerases (DNA pols) replicate genomic DNA or maintain the integrity of the host cell genome. These enzymes catalyze phosphoryl transfer reactions and incorporate deoxynucleotide monophosphate (dNMP) at the 3′-end of the growing chain by hydrolyzing deoxynucleotide triphosphates (dNTPs). Broadly, DNA pols can be divided into two groups: (i) replication DNA pols, and (ii) repair DNA pols. Replication DNA pols are required only once in the lifetime of a cell, whereas repair DNA pols are needed throughout the lifespan of a cell as mammalian cells are subject to ~70,000 lesions per day [1]. Generally, DNA pols have high fidelity and processivity and carry out template-dependent DNA synthesis. However, some DNA pols conduct error-prone DNA synthesis (low fidelity) and have low processivity, i.e., they conduct distributive DNA synthesis. A few DNA pols, such as terminal deoxynucleotidyl transferase (TdT), synthesize DNA in a template-independent manner [2].

Using sequence homology, DNA pols have been divided into eight major families: A, B, C, D, X, Y, RT (reverse transcriptase), and AEP (archaeo-eukaryotic primase) [3,4,5,6,7,8,9]. *E. coli* DNA polymerase I, the first discovered DNA pol [3,9], and DNA pols sharing sequence homology with *E. coli* DNA polymerases I, II, and III were grouped into A, B, and C Families, respectively [3,9]. *E. coli* DNA pol I (pol I) is one of the most studied Family A DNA polymerases. Additionally, a rigorous strategy was employed to reclassify Family A pols into 19 subfamilies [10]. Pol I has three functions: (i) 5′-3′ DNA synthesis, (ii) 3′-5′ exonuclease (proofreading) activity, and (iii) 5′-nuclease (also known as flap-endonuclease) activity. All these functions reside on the same polypeptide but on three structurally distinct domains [11]. Limited proteolysis of pol I results in two active fragments [11,12]. A large fragment of ~600 C-terminal residues known as Klenow fragment (KF), possesses both DNA synthesis and 3′-5′ exonuclease activities, and a smaller fragment (~300 amino acids) contains 5′-nuclease activity [12]. All known bacterial pol I homologs have high structural similarity and contain three distinct structural domains. However, some members do not have 3′-5′ exonuclease activity despite the presence of the structural domain [13,14]. The pol I orthologs in eukaryotes (except yeast) only have KF-equivalent proteins. For example, the catalytic subunit of mammalian polymerase γ (pol γ) has only the polymerase and proofreading domains [15]. Similarly, mammalian DNA polymerases θ and ν (pol θ and pol ν) have polymerase and proofreading domains and lack a 5′-nuclease domain [16,17,18,19,20,21]. While mammalian pol γ possesses proofreading activity, both pol θ and pol ν lack a conserved 3′-5′ exonuclease motif DxE therefore, they do not perform 3′-5′ exonuclease function [10]. Nonetheless, owing to structural homology with pol I, pol γ, pol θ, and pol ν have been conveniently referred to as Family A DNA pols [10].

Family A pols have been identified in almost all forms of biological entities, including viruses, plants, and parasitic organisms [10,22,23]. However, a pol I homologue in yeast has yet to be discovered. In recent years, there has been heightened interest among researchers in these enzymes due to their role in diseases such as cancer and malaria. Thus, inhibitors targeting the DNA synthesis function of human DNA pol θ [24,25,26,27] and *P. falciparum* apicoplast apPOL [28] have been reported. Development of competitive inhibitors with respect to dNTP substrate and allosteric inhibitors have been reported [24]. However, only one allosteric inhibitor has recently been cleared for Phase I/II clinical trials. The discovery of allosteric inhibitors targeting pol θ paves the way for developing compounds against bacterial Family A DNA polymerase, as these enzymes share high sequence and structural homology with the allosteric inhibitor binding pockets of pol θ and apPOL (discussed in the following sections), thereby providing opportunities for the development of a novel class of antibiotics. Due to low structural and sequence homology of DNA pol γ with pol θ, it appears unlikely that the same approach can be used to develop inhibitors against pol γ, even though pol γ has been associated with a variety of disorders [29]. 

## 2. Pol θ as a Drug-Development Target in HR-Deficient Cancers 

### 2.1. Human DNA pol θ

Human DNA pol θ, a multifunctional protein of 2590 amino acids (~290 kDa), is encoded by the *POLQ* gene. Pol θ has three distinct domains: (i) an 899 amino acid long N-terminal SF2 helicase domain, (ii) a ~771 amino acid long C-terminal DNA polymerase domain, and (iii) a 920 amino acid long central domain [30]. The structure of the N-terminal domain resembles an SF2 helicase and conducts NTPase activity, but the nucleic acid unwinding function is yet to be demonstrated [31]. The structure and function of the middle domain are not known. The C-terminal domain shares structural homology with the KF of *E. coli* DNA pol I [13,32]. The role of pol θ in maintaining genomic integrity and DNA repair has been extensively studied [30]. Some of the documented functions of pol θ include translesion DNA synthesis (TLS) past a variety of DNA lesions (bulky adducts, abasic sites, and interstrand crosslinks) [33], template-independent DNA synthesis [34], RNA-dependent DNA synthesis (reverse transcriptase activity), and RNA-dependent DNA repair [35], as well as pol θ-mediated end joining (TMEJ) [36,37]. Reports have shown that pol θ is overexpressed in breast, prostate, and lung cancer, and its inhibition can sensitize cancer cells to chemotherapy and radiotherapy [38]. A recent review by Wood and Doublie [30] details all the functions of pol θ. Therefore, we will only focus on recent drug discovery efforts targeting this enzyme. Additionally, we have restricted this report only to the polymerase domain of pol θ, despite the fact that the helicase domain is as significant as the polymerase domain for inhibitor development.

### 2.2. Human DNA pol θ as a Double-Strand Break (DSB) Repair Enzyme

In healthy cells, *BRCA1* and *BRCA2* serve as “tumor suppressor” genes. The proteins encoded by these genes (BRCA1 and BRCA2) repair double-stranded breaks (DSBs) [39]. Deficiency in DSB repair mediated by BRCA1 and BRCA2 can lead to the proliferation of cancer cells due to the accumulation of driver mutations. Alternative DNA repair pathways attempt to take over [40] in the event of compromised BRCA-mediated DSB repair [41]. All components of alternative DSB repair pathways are attractive anti-cancer drug-development targets, as cancer cells are prone to mutations and DSB generation forms the backbone of cancer therapies such as radiation-based therapies. One of the well-studied and successful proteins of the alternative DBS repair pathway is Poly (ADP-ribose) polymerase 1 (PARP1) [42]. Thus, a handful of anticancer drugs have been developed and approved against PARP [43,44] (Figure 1). Unfortunately, resistance to PARP inhibitors is an extremely common phenomenon in clinics, and more than 40% BRCA1/BRCA2-deficient patients fail to respond to PARP inhibitors [45] (Figure 1). Pol θ has been shown to conduct microhomology-mediated end-joining (MMEJ) (also known as theta-mediated end-joining or TMEJ) in the absence of functional BRCA1 and/or BRCA2 in cancer cells. Therefore, pol θ is considered an attractive anticancer drug development target for drug-resistant HR cancers. It should be noted that recent advances in the field indicate that BRCA mutants that exhibit DNA end-section have a heightened sensitivity to pol θ inhibitors [46]. Genomic expression data published in The Cancer Genome Atlas (TGCA) show that *POLQ* is significantly overexpressed in cancer when *BRCA1* or *BRCA2* are altered. The alterations, known as mutations, are shown in Figure 2. Therefore, the inhibition of pol θ is anticipated to be less toxic than conventional chemotherapy. Reports have shown that inhibition of pol θ using siRNA resulted in cancer cell death in vitro [41]. These results have triggered the identification of new therapeutics, such as small molecule inhibitors against pol θ.

### 2.3. Status of Drug Development against Human DNA pol θ

To date, four studies have reported inhibitors of pol θ that target its polymerase function [24,25,26,27]. Additionally, one study has reported a class of small molecules that inhibit the helicase function of pol θ [49]. A summary of pol θ inhibitors has been presented by Pismataro et al. [50]. Pol θ inhibitors targeting the polymerase domain can be divided into two classes: (i) competitive inhibitors (termed dxNTPs) with respect to dNTPs [24], and (ii) allosteric inhibitors [25,26,27]. Structurally related allosteric inhibitors ART558, ART812, and ART899 [25,27,51] bind in the ‘fingers’ subdomain of the polymerase domain and inhibit the polymerase function of pol θ (Figure 3), most likely via a mechanism similar to that of non-nucleoside RT inhibitors (NNRTIs). A small molecule inhibitor, GSK101 (IDE705), inhibits the ATPase function of pol θ, and an IND (Investigational New Drug) application has been cleared by the US FDA (August 2023). Other notable pol θ inhibitors are novobiocin (NVB) and RP-6685 [20,22,23]. NVB inhibits ATPase function [49], whereas RP-6685 is an allosteric polymerase inhibitor that binds in the ‘fingers’ sub domain of pol θ [26]. NVB is an antibiotic, which was discovered as an inhibitor of DNA gyrase [52]). It has been demonstrated that NVB and ART558 have high synergy with PARP inhibitors and decrease IC_50_ values significantly in HR-deficient cells [51]. RP-6685 was identified in 2022, and in mouse models bearing BRCA-deficient tumors, treatment with RP-6685 resulted in decreased tumor volume, although a slight increase was observed after the 21st day of treatment [20,24]. The structures of these inhibitors and their target sites are shown in Table 1.

## 3. Apicoplast DNA Polymerase (apPOL) as an Antimalaria Target 

### 3.1. Status of Drug Development against apPOL

According to the World Malaria Report 2022 (https://www.who.int/teams/global-malaria-programme/reports/world-malaria-report-2022, accessed on 25 February 2024), malaria kills more than half a million people every year. Malaria is caused by the parasites of the *Plasmodium* genus of Apicomplexa phylum [53]. The majority of organisms belonging to the Apicomplexa phylum contain an apicoplast that is evolutionarily related to the chloroplast [53,54,55,56]. The apicoplast participates in several metabolic pathways, and it is essential for the survival of the parasite [57]. A Family A DNA polymerase known as apPOL is an integral part of the replisome that copies the 35 kb genome of the apicoplast [56,58] (Figure 4). Phylogenetic analyses have shown that apPOL has low sequence homology (23%) with human DNA pol θ. However, the crystal structure has shown that apPOL has a canonical KF fold (PDB files 7SXL and 7SXQ) [28], and the polymerase domain of apPOL superimposes on the polymerase domain of pol θ (PDB file 8E23) with a root mean square deviation (RMSD) of ~1.5 Å, and with ~1.9 Å with the Klenow fragment equivalent of *Bacillus stearothermophilus* despite low (25%) sequence homology [59]. 

### 3.2. Apicoplast as an Antimalaria Drug Target

The majority of drug development against apPOL has been conducted by Nelson and colleagues. Screening of compounds from the Open Access Malaria Box drug collection resulted in the identification of compound MMV666123 [54]. Further attempts at structure–activity relationship (SAR) studies of MMV66123 (termed compound **5a**) did not provide a compound with better potency [28]. Attempts to solve the crystal structure of apPOL in complex with **5a** did not succeed. However, mutational data showed that compound **5a** binds close to W512. The equivalent residues in KF and pol θ are Y801 and F2426, respectively. In the crystal structure of pol θ in complex with an RP-6685 analog (PDB file 8E24) [26], residue F2426 is not within interacting distance of the compound. Therefore, F2426 may serve as the ‘gate’, as suggested by Chheda et al., for W512 [28]. It is possible that RP6685 and related compounds bind in the ‘fingers’ subdomain of apPOL at the RP-6685 equivalent site involving the residues of the O, O1, and O2 helices, as the pol θ residues interacting with the RP-6685 analog are well-conserved across Family A DNA pols (Section 4.4). Other antimalarial compounds under development include pyrrole-hydroxybutenolide hybrids [60]. It is also possible that residues W512 and I422 of apPOL may act similarly to E138 of the p51 subunit K101 and the p66 subunit of HIV-1 RT, which serve as a ‘gate’ for the binding of 2nd-generation non-nucleoside inhibitors (NNRTI) (PDB files 3MEE and 2ZD1) [61,62,63].

## 4. Pol I as a Target for Combating Antimicrobial Resistance 

### 4.1. Antimicrobial Resistance: A Global Health Concern

Antimicrobial resistance (AMR) is a serious public health concern. According to the Antibiotic Resistance Threats Report 2019 from the CDC, ~2.8 million AMR infections occur each year, causing ~36,000 deaths [64]. Globally, these numbers are significantly higher. In 2019, ~4.3 million deaths were associated with bacterial antibiotic resistance, of which 1.27 million deaths were directly attributed to bacterial AMR [65]. Closely connected humans, animals, and environmental habitats contribute to the emergence, evolution, and spread of bacterial AMR [66]. Thus, major multidrug-resistant (MDR) organisms have been recovered from humans, animals, and the environmental habitat [66]. The drivers of bacterial AMR are manifold, but excessive antibiotic use and misuse in humans and animals are two major drivers of AMR [67,68,69]. Other socioeconomic factors, such as the paucity of clean water, have led to the development of resistance to almost all currently used antibiotics [64]. All of the above-mentioned factors have sparked renewed interest among researchers to investigate alternatives to currently prescribed antibiotics as treatment options against drug-resistant bacterial infections [70,71].

### 4.2. Family a Polymerases as a Novel Antibacterial Target

The pol I group of enzymes participates in Okazaki fragment maturation during bacterial genome replication [11]. This group of enzymes is also crucial for cell survival following DNA damage. Due to their ability of nick-translation, pol I enzymes function as the effectors for the ligase-mediated sealing of single-stranded nicks. A nick-sealing property does not appear to be present in other DNA polymerases [72]. Mutations in *E. coli* pol I have been shown to confer high sensitivity to UV radiation and methyl methanesulphonate [73]. Pathogenically, pol I has been shown to be essential for the viability of, otherwise impaired, *dnaN159 E. coli* [74], and the survival of *H. pylori* [75]. In *S. Typhimurium LT2*, pol I is required for the uses of ethanolamine, 1,2-propanediol, or propionate as the carbon and energy source [76]. These examples underscore the importance of bacterial DNA pol I as drug targets.

### 4.3. Pol I Inhibitors Acting through a Mechanism Analogous to NNRTI Inhibition of HIV-1 RT

The structure of KF (the first DNA polymerase structure) [13] showed that the polymerase domain of these enzymes consists of subdomains that resemble a half-open right hand, leading to the nomenclature of these domains as the ‘thumb’, fingers’, and the ‘palm’. Subsequent structures of pol I enzymes showed that the ‘thumb’ and ‘fingers’ subdomains undergo substantial conformational changes to form a catalytically competent ternary complex consisting of template-primer and dNTP substrate [77]. A comparison of HIV-1 RT in complex with NNRTI nevirapine [78] and HIV-1 RT in complex with DNA and dNTP [79] showed that the binding of NNRTI restricts the movement of the ‘thumb’ subdomain among other conformational changes called ‘molecular arthritis’ [78,80]. Reported compounds that bind in the ‘fingers’ subdomain of pol θ appear to inhibit it via a mechanism that is analogous to NNIRT inhibition of HIV-1 RT [81].

### 4.4. Comparison of Allosteric Inhibitor Binding Sites in Pol I Enzymes from Diverse Species

As reported for bacterial pol I [77], the ‘fingers’ subdomain of pol θ undergoes substantial conformational change to a closed conformation (Figure 5). A comparison of pol θ in complex with template-primer (t/p) and dideoxyguanosine 5′-triphosphate (ddGTP) (PDB file 8E24) [26] and pol θ bound to RNA/DNA (PDB file 6XBU) [35] shows that the O-helix significantly rotates towards the ‘palm’ subdomain (Figure 5). The binding of the allosteric inhibitor RP-6685 analog to the ternary complex of pol θ, consisting of t/p and ddGTP (PDB file 8E24), locks the O-helix in its closed conformation (Figure 5) [26] and restricts the conformational change of the O-helix, since a part of the inhibitor occupies the position where the O-helix is expected to be during pyrophosphate release and the translocation steps of DNA polymerase catalysis (in the open fingers conformation) [82,83] (Figure 5).

To assess if the pol θ allosteric inhibitor binding pocket is also present in other pol I enzymes at the topological position, we generated structures of pol I enzymes using pol θ in complex with p/t, ddGTP, and hit compound **5** (PDB file 7ZX0) [27] as a template with the Modeller program [84]. We then docked compound **5** using the ‘Glide’ docking program of Schrödinger Suite (Schrodinger LLC, NY) in a pocket formed by N, O, O1, and O2-helices of pol I enzymes from 12 bacterial species, apPOL, and human pol ν. Compound **5** was docked with a high docking score (Glide score < −7) in the ‘fingers’ subdomain of all bacterial pol I enzymes, as well as in the crystal structures of apPOL [28] and pol ν (PDB file 4XVK) [85]. The best docking pose of **5** in *E. coli* pol I is shown in Figure 6. It is clear from this figure that the majority of compound **5** interacting amino acid sidechains are conserved between pol θ and *E. coli* pol I (8 out of 11) (Figure 6). Additionally, compound **5** interacting residues are conserved across the pol I family, suggesting that structure–activity relationship approaches can be used to design species-specific allosteric inhibitors of the pol I family of enzymes.

## 5. Conclusions

In response to the challenge of mutations under pressure from drugs to treat cancers, malaria, and microbes, novel solutions are continuously required and investigated. These solutions include novel targets and the application of novel drugs to different targets. For example, cancers with HR deficiencies can be targeted by developing inhibitors against pol θ; the inhibitors of pol θ can also act against opportunistic infections such bacterial and parasitic infections due to the structural homology seen in Family A polymerases. Structure–activity relationship strategies can be applied to develop disease-specific inhibitors. Furthermore, compounds against unexplored targets such as bacterial DNA pol I can be expected to target multi-drug resistant bacteria and thereby help in combating AMR.

## Figures and Tables

**Figure 1 biology-13-00204-f001:**
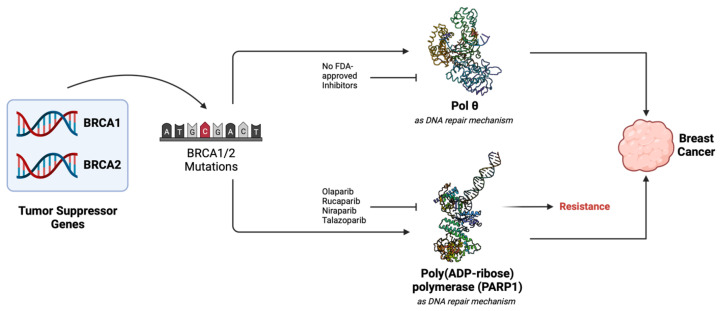
Role of BRCA1/2 proteins, PARP, and pol θ in cancers. Mutations or alterations in BRCA1/2 genes result in compromised homologous recombination (HR) in cancer cells. As a result of genomic instability, cancer cells start proliferating. These cells rely on the PARP1 gene for HR, which is overexpressed in cancer cells. Therefore, inhibitors targeting PARP have been designed and approved to suppress the growth of cancer. However, ~40% of patients develop resistance to PARP inhibitors. BRCA-deficient cells also rely on pol θ for their continued growth. While PARP1 is generally considered a major target of PARP inhibitors (PARPi) due to the structural similarity within the NAD-binding domain; some PARPi also inhibit PARP2 and PARP3 [43]. Thus, pol θ is an important target for inhibitors to prevent the growth of cancer cells. PDB file 4DQY [47] for PARP1 and PDB file 8E23 [26] for pol θ was used to generate these figures.

**Figure 2 biology-13-00204-f002:**
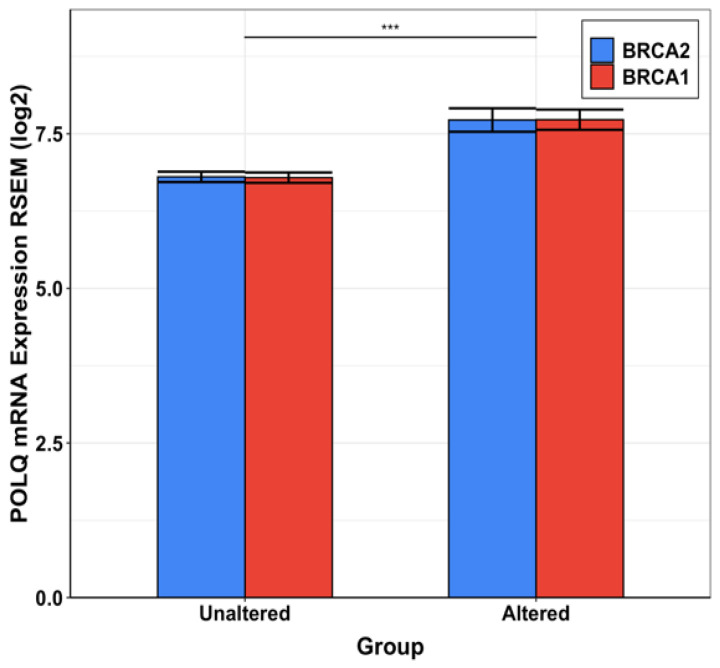
Expression of *POLQ* in normal cells and the cells containing altered (or mutant) BRCA1/2. This figure shows the difference in *POLQ* expression between normal cells and cells with BRCA mutations. Statistically significant higher *POLQ* expression is noted in normal cells compared to BRCA mutant cells. *** Based on a statistical *T*-test, the difference in *POLQ* mRNA expression between the two groups, unaltered BRCA and altered BRCA, is statistically significant (*p* < 0.001). Data used to generate this figure were obtained from CBioPortal [48].

**Figure 3 biology-13-00204-f003:**
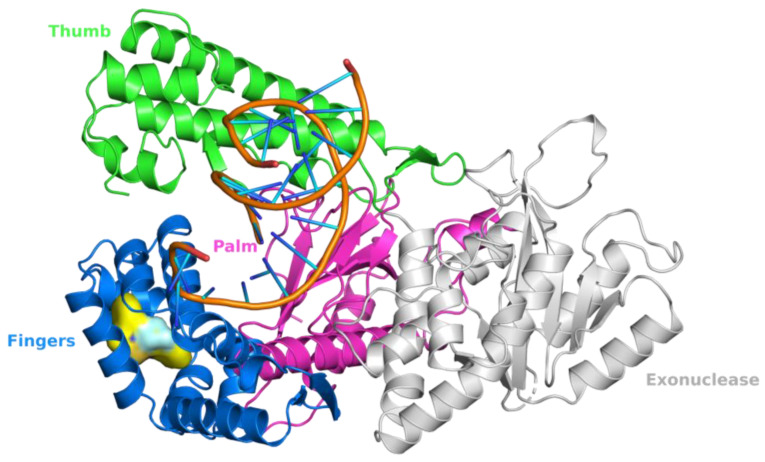
Structure of human DNA pol θ in complex with an allosteric inhibitor. This figure shows that compound **14** (a hit that led to the development of RP-6685), shown in surface representation [26] (PDB file 8E23), binds in the ‘fingers’ subdomain of human DNA pol θ. The ‘thumb’, ‘palm’, ‘fingers’, and 3′-5′ exonuclease domain of pol θ are shown as green, magenta, blue, and gray ribbons, respectively.

**Figure 4 biology-13-00204-f004:**
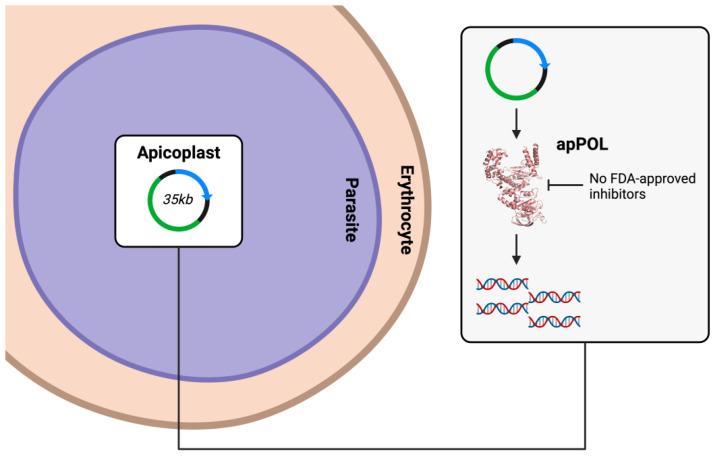
Enzyme apPOL as an antimalaria target. The apicoplast is an essential organelle in parasites with a genome of ~35 kb. This genome is copied by a replication machinery that requires apPOL. Thus, the inhibition of apPOL is expected to suppress the replication of the apicoplast genome. Currently, there are no approved inhibitors of apPOL. PDB file 7SXQ [28] was used to generate the ribbon diagram of apPOL.

**Figure 5 biology-13-00204-f005:**
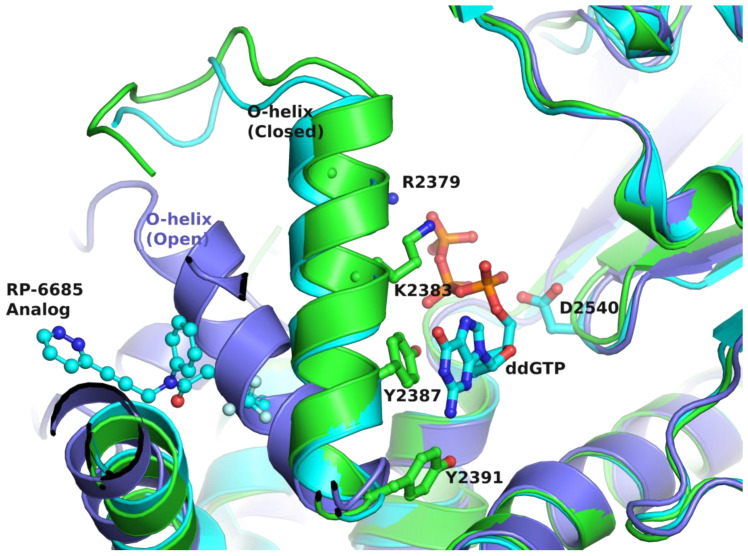
Mechanism of allosteric inhibition of pol θ. Superposition of complexes of pol θ-RNA/DNA (PDB file 6XBU; [35]) (violet ribbons), pol θ-t/p-ddGTP-RP6685 analog (PDB file 8E24; [26]) (green ribbons), and pol θ-t/p-ddGTP (PDB file 4X0Q; [32]) (cyan ribbons), providing a possible mechanism of inhibition of the allosteric inhibitor bound at the ‘fingers’ subdomain. The RP6685 analog is shown as ball-and-sticks (cyan carbons). O-helix residues known to participate in dNTP binding and pyrophosphate release, ddGTP, and one of the three catalytic site carboxylates (green carbons) are rendered as balls-and-sticks, serving as references to the inhibitor binding site. The atoms nitrogen, oxygen, phosphorus, and fluorine are colored blue, red, orange and light cyan, respectively. This figure illustrates how the movement of the O-helix creates an inhibitor binding pocket and how the bound inhibitor restricts conformational changes of the O-helix.

**Figure 6 biology-13-00204-f006:**
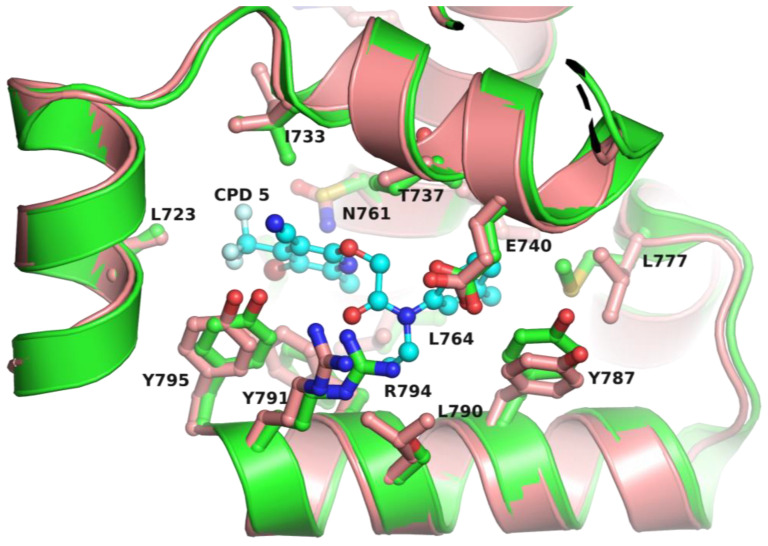
Representative figure showing the conservation of inhibitor binding site residues between Klenow Fragment and pol θ. This figure shows the superposition of pol θ (green ribbons) and a homology-based modeled KF using PDB file 7ZX0 as the template by the Modeller program [84] (pink ribbons) bound to hit compound **5** (CPD 5) (PDB file 7ZX0) of ART558 [27]; the amino acid residues that interact with the compound are represented in ball-and-stick form (pink carbons for *E. coli* pol I and green carbons for pol θ). The atoms nitrogen, oxygen, sulfur, and fluorine are colored blue, red, yellow, and light cyan, respectively. The numbering of amino acid residues corresponds to *E. coli* DNA pol I.

**Table 1 biology-13-00204-t001:** Inhibitors under development against the pol I family of enzymes.

Compound Name	Structure	Target
ART558 ^1^	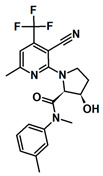	Pol θ Polymerase Function
RP-6685	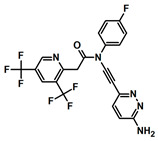	Pol θ Polymerase Function
GSK101 (IDE705) ^2^	Undisclosed	Pol θ Helicase Function
Novobiocin ^2^	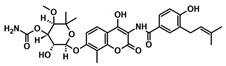	Pol θ Helicase Function
MMV666123	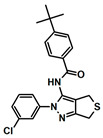	apPOL Polymerase Function

^1^ Compound ART558 is a member of a family of compounds (ART558, ART812, ART889). For representative purposes, only ART558 is shown. ^2^ These compounds target the helicase domain of pol θ and not the polymerase domain. These compounds are included to provide an update on pol θ inhibitor development.

## Data Availability

Not applicable.

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
