# Peer review of "Inhibitors against DNA Polymerase I Family of Enzymes: Novel Targets and Opportunities"

_biology, 2024, doi:10.3390/biology13040204_

Round 1
Reviewer 1 Report
Comments and Suggestions for Authors
Review on the manuscript “Inhibitors against DNA polymerase I family of enzymes: Novel 2 targets and opportunities”.
Manuscript describes DNA polymerase I family of enzymes and its inhibition. This class of enzymes is of great interest due to its roles in cancer and microbial infections. Compounds targeting apicoplast apPOL possess antimalarial activity. Thus reviewing this field of research has significant science interest.
Authors give a short review on DNA polymerase I family of enzymes and its inhibitors and following therapeutic effects. Authors cited huge amount of papers in this field and no excessive self-citation was detected.
Manuscript could be published in Biology after following corrections will be made:
First of all, authors should add country of their origin to the affiliation. One can guess that they are from the USA but it should be added anyway. Moreover full name of the state should be given.
Manuscript gives a short review on the inhibitors of certain DNA polymerases by small molecules. However no structures of such molecules were given in the manuscript.
Please add a figure with chemical structures of small molecules mentioned in the text. This will increase appearance on the manuscript especially by medicinal chemists and more likely manuscript will be referred by search engines like Reaxys and SciFinder giving more citations.
Author Response
Reviewer 1:
- First of all, authors should add country of their origin to the affiliation. One can guess that they are from the USA but it should be added anyway. Moreover, full name of the state should be given.
We thank the reviewer for the suggestion. We have modified the affiliations accordingly.
- Manuscript gives a short review on the inhibitors of certain DNA polymerases by small molecules. However, no structures of such molecules were given in the manuscript. Please add a figure with chemical structures of small molecules mentioned in the text. This will increase appearance on the manuscript especially by medicinal chemists and more likely manuscript will be referred by search engines like Reaxys and SciFinder giving more citations.
We thank the reviewer for the insightful suggestion. We have now added a table (Table 1) that contains not only the structures of the inhibitors but also their targets.
Reviewer 2 Report
Comments and Suggestions for Authors
07.03.2024
A review to evaluate its suitability for publication Type of manuscript:
Title: Inhibitors against DNA polymerase I family of enzymes: Novel targets and opportunities
Authors: Saathvik Kannan, Samuel W. Gillespie, W. L. Picking, W. D. Picking, Christian L. Lorson, Kamal Singh
This review is devoted to a detailed description of the functioning of a variety of DNA polymerases, in particular, the γ, θ families, which play a role in the development of various pathologies. Knowledge of the mechanism of action of some A-family polymerases, including the use of artificial intelligence, makes it possible to develop drugs - inhibitors of DNA polymerase enzymes (A-family), as a means of combating pathologies.
There are the comments:
1. Line 12-25: Simple Summary requires revision and changes, as the form in which it is now presented is more like a reference (encyclopedic) information. There is a need to describe those new approaches, including molecular modeling, which significantly distinguish this Summary from numerous other previously published ones [1. Faraoni I, Graziani G. Role of BRCA Mutations in Cancer Treatment with Poly(ADP-ribose) Polymerase (PARP) Inhibitors. Cancers (Basel). 2018 Dec 4;10(12):487. 2. Konecny GE, Kristeleit RS. PARP inhibitors for BRCA1/2-mutated and sporadic ovarian cancer: current practice and future directions. Br J Cancer. 2016 Nov 8;115(10):1157-1173.3. Zatreanu, D., Robinson, H.M.R., Alkhatib, O. et al. Polθ inhibitors elicit BRCA-gene synthetic lethality and target PARP inhibitor resistance. Nat Commun 12, 3636 (2021)].
2. Line 98: Figure 1: The diagram shows PARP1 polymerase as a "sensor" of DNA damage. What is the role in induces synthetic lethality of other varieties of polymerases - PARP2 and PARP3?
3. Line 152: Is Figure 2 an authorship or a borrowing? The source of data on the expression of POLQ in normal cells and the cells containing altered cells should be indicated.
4. Line 177: Figure 3:
4.1. Whose authorship of this docking complex?
4.2. Which software products (AutoDock4, AutoDock4(Zn), AutoDock Vina, DOCK, MpSDockZn, PLANTS и Surflex-Dock) were used to perform docking of pol θ in complex with allosteric inhibitor?
4.3. What is compound 5?
5. How can the success of the docking exercise be assessed using quantitative criteria?
6. Line 197: Section 3.2. Apicoplast as antimalaria drug target needs to be supported by a figure - schematic or docking diagram
7. Line 215: Section 4 Pol I as a target for combating antimicrobial resistance: explain what is the causal relationship between the paucity of clean water and the development of resistance to almost all currently used antibiotics?
8. Figures 4 and 5: Authorship of this docking model? What software products were used for docking?
Respectfully, reviewer
Author Response
Reviewer 2:
- Line 12-25: Simple Summaryrequires revision and changes, as the form in which it is now presented is more like a reference (encyclopedic) information. There is a need to describe those new approaches, including molecular modeling, which significantly distinguish this Summary from numerous other previously published ones [1. Faraoni I, Graziani G. Role of BRCA Mutations in Cancer Treatment with Poly(ADP-ribose) Polymerase (PARP) Inhibitors. Cancers (Basel). 2018 Dec 4;10(12):487. Konecny GE, Kristeleit RS. PARP inhibitors for BRCA1/2-mutated and sporadic ovarian cancer: current practice and future directions. Br J Cancer. 2016 Nov 8;115(10):1157-1173.3. Zatreanu, D., Robinson, H.M.R., Alkhatib, O. et al. Polθ inhibitors elicit BRCA-gene synthetic lethality and target PARP inhibitor resistance. Nat Commun 12, 3636 (2021)].
The reviewer’s suggestions are excellant. However, the instructions for the 'Summary' section of ‘Biology (Basel)’ clearly state that this section should be written for the lay audience. This was the only reason, we restricted ourselves as to encyclopedic information. Nonetheless, as the reviewer suggested, we have added additional statements at the end of the summary. In doing so, we had to edit the original ‘Summary’ section to contain this section to 200 words as per the journal’s instructions.
Line 98: Figure 1: The diagram shows PARP1 polymerase as a "sensor" of DNA damage. What is the role in induces synthetic lethality of other varieties of polymerases - PARP2 and PARP3?
We agree with the reviewer. While PARP1 is generally considered the major target of PARP inhibitors (PARPi) due to the structural similarity of the NAD-binding domain of some of the PARP family members, some PARPi also inhibit PARP2 and PARP3. We have now included statements addressing this concern.
- Line 152: Is Figure 2 an authorship or a borrowing? The source of data on the expression of POLQ in normal cells and the cells containing altered cells should be indicated.
This figure was made based on data from the Tumor Cancer Genome Atlas (TCGA) and CBioPortal. We have now included data source and the citations in the figure legend, and in the 'References' section .
- Line 177: Figure 3: Whose authorship of this docking complex? Which software products (AutoDock4, AutoDock4(Zn), AutoDock Vina, DOCK, MpSDockZn, PLANTS и Surflex-Dock) were used to perform docking of pol θ in complex with allosteric inhibitor? What is compound 5?
We thank the reviewer for identifying this error regarding ‘Compound 5’; we have modified the legend to accurately reflect the inhibitor shown in the figure (i.e., compound 14). The inhibitor shown in Figure 3 is not a result of a docking—it is a determined by X-ray crystallography (PDB file 8E23). The X-ray crystallography data showed that the hit compound that led to the development of RP-6685 binds at an allosteric site which is generated by a conformational change in the ‘fingers’ subdomain of polymerase θ.
- How can the success of the docking exercise be assessed using quantitative criteria?
As stated above (in 3), we did not use docking to generate Figure 3. The docking was carried out to dock compound 5 from PDB file 7ZX0 in the ‘fingers’ subdomain of other Family A polymerases. We used ‘Glide’ of Schrödinger Suite (Schrödinger LLC, NY, USA). We would like to point out here that the docking was performed only to evaluate if compounds that bind pol θ have potential to bind other Family A polymerases, and if a similar binding pocket exists in these polymerases. As we expected, compound 5 could be docked in all Family A polymerases that we included with a Glide score < -7, and all these polymerases contain fairly conserved allosteric compound 5 binding pocket. Additionally, we only showed the binding of compound 5 in the Klenow Fragment of E. coli DNA pol I in Figure 5. We have now added the quantitative metric for docking in Figure 5 using the Glide Score as a metric.
- Line 197: Section 3.2. Apicoplast as antimalaria drug targetneeds to be supported by a figure - schematic or docking diagram.
We thank the reviewer for the suggestion. We have now included a detailed figure for the malarial target justification.
- Line 215: Section 4 Pol I as a target for combating antimicrobial resistance: explain what is the causal relationship between the paucity of clean water and the development of resistance to almost all currently used antibiotics?
We thank the reviewer for the insightful suggestion, and it is an excellent question. However, the focus of this review is not on how the resistance is developed in bacteria in the absence of clean water. Instead, the review is focused on identifying novel targets as opportunities to combat this resistance. We made this statement based on statements/studies by the U.S. Center for Disease Control and Prevention (reference number 64), which provides substantial evidence and details regarding the causal link. As such, clean water is one of many major factors that are responsible for the development of antimicrobial resistance. The other factors include extensive use and misuse of antibiotics (please see reference # 64 for details).
- Figures 4 and 5: Authorship of this docking model? What software products were used for docking?
The inhibitor shown in original Figure 4 (now Figure 5) is not a result of a docking. Instead, it was determined by X-ray crystallography. In fact, all structures used in this figure have been determined by X-ray crystallography. Three PDB files 4X0Q, 6 XBU and 8E24 were used to generate this figure. The ‘palm’ subdomain was used to superpose three structures. The compound shown in this Figure is an analog of RP6685. This figure demonstrates that movement of fingers in the enzyme/template-primer/dNTP ternary complex generates an allosteric inhibitor binding binding pocket. It appears that the binding of the inhibitor restrict the conformational change of the fingers subdomain and thereby inhibits the polymerase function of pol θ. Software PyMol was used to generate an in-house figure. We have updated these details in the revised manuscript. Original Figure 5 (now Figure 6) used the ‘Glide’ docking methodology specified earlier. The docking was performed only to evaluate if compounds that bind pol θ have potential to bind other Family A polymerases, and if a similar binding pocket exists in these polymerases. As we expected, compound 5 could be docked in all Family A polymerases that we included with a Glide score < -7, and all these polymerases contain fairly conserved allosteric compound 5 binding pocket. Additionally, we only showed the binding of compound 5 in the Klenow Fragment of E. coli DNA pol I in Figure 5. We have now added the quantitative metric for docking in Figure 5 using the Glide Score as metric as well.
Round 2
Reviewer 1 Report
Comments and Suggestions for Authors
Authors have made all necessary corrections.
Author Response
Thank you for your time and insightful suggestions.
Reviewer 2 Report
Comments and Suggestions for Authors I’ve just looked through the revised text and author’s answers,- yes, I absolutely satisfy it! The authors have changed the manuscript very carefully.
Author Response

(The authors gave the same response as above.)
